# Research on the Impact of Pro-Environment Game and Guilt on Environmentally Sustainable Behaviour

**DOI:** 10.3390/ijerph192013406

**Published:** 2022-10-17

**Authors:** Jiaxing Chen, Guangling Zhang, Qinfang Hu

**Affiliations:** 1School of Economics and Management, Wuhan University, Wuhan 430074, China; 2School of Economics and Trade, Hunan University of Technology, Zhuzhou 412007, China

**Keywords:** guilt, pro-environment behaviour, pro-social behaviour, game strategies

## Abstract

Game strategies are widely used by companies to attract users and increase their stickiness. At the same time, the protection of the ecological environment is also an important expression of corporate social responsibility. This paper explores the integration of social responsibility with gaming strategies from the psychological perspective of game withdrawal, and explores the incorporation of social responsibility as an element in gamification design to reduce user withdrawal behaviour, thereby increasing individual’s environmentally sustainable behaviour. We evidenced our hypothesis through two studies. Study one proved our hypothesis by recruiting 106 university undergraduates (from Wuhan University, mean age 20, of whom 47 were female and 59 were male) to prove our hypothesis by recalling previous experiences with different types of games. Study two further tested our hypothesis by manipulating participants’ guilt through randomly recruiting 196 participants (mean age 35, of whom 88 were female and 108 were male, 35 of them were students, 107 were office workers and 54 were from other sectors) from different industries through the questionnaire research website Credamo. The findings show that incorporating social responsibility elements into the design of games can make users engage in pro-social behaviour while playing the game, and the guilt that users feel because of the game will be compensated by pro-social behaviour, thus reducing the game frequency and duration and improving the intent of pro-social behaviour. At the same time, players’ self-control moderates the effect of guilt on game play volume under a socially responsible gamification design.

## 1. Introduction

Online games are currently an important means of attracting users to platforms [1], and many platform companies design various types of game features on their platforms to attract users; for example, WeChat as a social platform has also designed a popular airplane game on its login screen [2]. The integration of social responsibility elements into the gamification design is a major exploration of the gamification strategy of platform companies [3]. For example, the Ant Forest game on the Alipay platform incorporates the greening act of protecting the environment into the platform’s mini-games, and users indirectly plant real trees in the desert area while playing the game [4]. a report by the State Ministry of Environmental Protection in August 2019 also shows that Ant Forest has reduced carbon emissions by 7.92 million tons in three years [5]. There are also, for example, WeChat Donate Steps on the WeChat platform, Ali’s Walk to Donate, etc. [6]. This combination of pro-social behaviour, such as environmental protection and donation with gamification, allows the platform to attract users and satisfy corporate social responsibility obligations, while also prompting consumers to become more sustainable in their pro-social behaviour by playing games [7].

While gamification can provide players with a sense of enjoyment, control, competence, and other positive effects [8], it can also have negative effects [9]. For example, gamers may be blamed by family members for playing the game, which may lead to family discord, or they may waste time playing the game, which may delay their normal work, school, or life schedule. This guilt becomes a disincentive to play, which may lead to withdrawal behaviour [10]. At the same time, players may also abandon the game due to a sense of responsibility towards others or themselves when they become aware of other meaningful activities [11].

Players’ gameplay withdrawal behaviours influence the platform’s gamification strategy [12]. In order to mitigate the negative effects of lack of responsibility and guilt when playing games, platforms are designed in gamification to allow players to enjoy both the positive effects of playing games, but also to include an element of social responsibility, allowing players to play games while also engaging in pro-social behaviour, which reduces game withdrawal behaviour because pro-social behaviour reduces guilt. This design also makes playing the game a pro-social act, which not only increases the sense of responsibility when playing the game but also motivates players to engage in more pro-social behaviour due to the positive impact of the game.

Existing research on gamification has mainly investigated the design elements and mechanisms of gamification from the perspective of games that attract players and give them a sense of control, competence, hedonism, connectedness, and flow experiences [3,13,14,15,16,17]. Some of the literature that applies game strategies to the environmental field focuses on the role of game strategies in terms of their effectiveness in enhancing environmental strategies [18]. Some studies have shown that game mechanics and elements can promote eco-friendly behaviour from a habit-forming perspective, thus making eco-friendly behaviour a green lifestyle [19]. Other studies have explored the motivations of users to gain a social reputation through environmental gaming behaviour from the perspective of social identity [20]. Few studies have combined the negative psychological and behavioural effects arising from the gamification experience with pro-social behaviours and considered their joint effects on players’ psychology and behaviour. This paper, therefore, investigates the effects of incorporating socially responsible elements into the design of gamification on players’ guilt and game duration, based on a psychological mechanics of game withdrawal perspective.

## 2. Conceptual Framework and Research Hypothesis

### 2.1. Game

Games generally refer to computer and mobile games, and when playing games, people typically experience feelings of control, competence, enjoyment, immersion, or flow, all of which are characteristic of intrinsically motivated human behaviour [21]. When starting a game, players accept the serendipity of the result; yet, regardless of the outcome, the process is often enjoyable.

Self-determination theory is generally used to explain the underlying motivation to play games [22], i.e., the idea that humans are essentially motivated either intrinsically or externally, depending on whether the activity is motivated by the purpose of the activity itself or by reasons external to the activity, where for intrinsically motivated behaviours it may stem from a motivational need for competence, autonomy, and relationality. For example, games can help players to de-stress and relax after work while providing a sense of control by controlling the characters in the game. Games can also provide challenge and a sense of competence, connection, and immersion. For example, branch points, badges, quests, leaderboards, and timers can satisfy a sense of competence, social networking features, group voting, and team games satisfy a sense of social connection, and virtual identities, in-game rewards, and role-playing provide a sense of immersion. The purposes for which people play games can be divided into three categories: firstly, gaming as a lifestyle; secondly, gaming communities as a subculture; and thirdly, gaming as a purposeful activity, e.g., gaming can provide opportunities to relax, exert control, be challenged, and achieve goals [10].

Although gaming has many advantages, it also has several negative effects. For example, the gaming community is seen as a subculture that is often misunderstood by the dominant culture [23], and the partners of gamers may see gaming as an activity only for children and not for adults, which can put pressure on gamers. It is also difficult for outsiders to understand gaming as a hobby; they see it as a waste of time and can cause gamers to experience a state of mind where they both want to play and feel guilty about it. Studies have also explored the negative societal attitudes and psychological impact on gamers in terms of the stigma of gaming [24].

The negative effects of games can affect game usage. Some scholars have categorized the factors that influence the amount of gaming as mainly personal, interpersonal, and environmental influences [10]. Personal influences include the player’s sense of responsibility for other tasks and roles outside of the game, planning or scheduling, and meeting the need for self-care. Sense of responsibility comes from being responsible for oneself (e.g., earning money to pay bills/buy things) or for others (e.g., being a caregiver for a family member) and refers to players giving up the game when other meaningful activities are available. Planning or scheduling refers to having other things to do. Satisfying self-care needs refers to when the player uses the game as a way to cope with negative emotions or negative life events (such as the death of a family member or a break-up with a partner), the game provides a way to escape reality, but using the game as a way to cope only temporarily relieves negative emotions and does not solve the problem the gamer is facing, and after playing may feel that it was a waste of time or may feel worse, all of which can cause gamers to leave the game. Interpersonal influences, including the influence of cohabitants (family members and partners), friends, social apps/websites, colleagues, teammates, etc., on the amount of play. For example, living with parents may make the gamer feel guilty and thus forced to stay away from the game. Environmental influences refer to whether the features of the game itself are appealing, whether the gaming device supports it, whether environmental conditions are supportive, and some restrictions on access to the game.

Cognitive preoccupation refers to a compulsive pattern of thinking when using the Internet, such as ‘I can’t stop thinking about the Internet’ or ‘I can’t stop thinking about what’s happening online when I’m offline [25,26]. When online gaming is more engaging, it can lead to excessive gaming due to cognitive addiction [12,27], which can have many negative psychological effects and physical and psychological problems, such as psychosocial problems, i.e., addiction to gaming without real-life relationships, carelessness, aggressive behaviour and hostility, stress, inappropriate coping styles, reduced academic performance, reduced verbal memory skills. In 2018 the World Health Organisation (WHO) is also in the Classification of Diseases ICD11, defined gaming disorder (Gaming disorder and hazardous gaming are defined as diseases. Hazardous gaming behaviour’ refers to a pattern of gaming, whether online or offline, that significantly increases the risk of harmful physical and mental health consequences for the individual or others around them, either because of the frequency of playing, the time spent on these activities, the neglect of other more important things, the risky behaviours associated with gaming disorders are considered to be mental disorders of the same nature as drug addiction, tobacco addiction, gambling addiction, etc. [12]. Research has shown significant associations between people’s internet use, mental health, and negative outcomes at home and work [27].

### 2.2. Game Strategy

The game strategy includes using games or gamification to attract users. One nature of playing games is the self-purpose nature of the activity, as well as the engagement and enjoyment of the activity. Gamification, on the other hand, is about capturing, exploiting, and implementing these properties of games through gamification techniques for application to more instrumental purpose environments, enhancing the functionality that can be achieved by platform software by designing game-like experiential elements that engage users by providing them with a sense of pleasure, flow experience, autonomy, control, competence, etc., and thus influencing user behaviour [28,29], which consists of three main elements: availability, psychological experience, and behavioural outcomes.

Game affordance refers to the various elements and mechanisms that make up the game and help induce a gaming experience within the system [30], e.g., points, badges, quests, leaderboards, timers, etc., can satisfy a sense of competence, social networking features, group voting, team play, etc. satisfy a sense of social connection, virtual identities, in-game rewards, role-playing, etc. provide immersion, some non-digital elements such as location data, motion tracking, real-world interactive objects, and other elements such as virtual reality, virtual currency, etc. Psychological outcomes refer to psychological experiences such as competence, autonomy, and relatedness, or feelings of enjoyment and engagement, such as evaluations of the game or gamification, personal emotions, perceptions, level of effort spent using, the difficulty of the game characters, attitudes, perceptions of others, etc., which games and gamification are often thought to facilitate. The behavioural outcomes of gamification are the behaviours and activities that are supported using gamification systems, such as sustained or increased physical activity in the context of exercise gamification or better learning outcomes in the context of educational gamification [28].

### 2.3. Social Responsibility, Guilt, and Ethical License

Social responsibility refers to a person’s sense of responsibility to society, where people can feel morally or traditionally responsible for society, and it is a predictor of that person’s ecological behaviour. Moral responsibility depends on a person’s self-attributed responsibility (i.e., intentional stress response judgments) and guilt. Traditional responsibility depends on a person’s awareness of social expectations and his willingness to meet those expectations [31,32]. Pro-environmental behaviour is a mix of self-interest (e.g., pursuing a strategy to minimize one’s own health risks) and concern for other people, future generations, other species or the ecosystem as a whole (e.g., preventing air pollution that may pose a risk to the health of others or to the global climate). Researchers who view pro-environmental behaviour as primarily pro-socially motivated often use the norm-activation model. The basic premise of the norm-activation model is that moral or personal norms are a direct determinant of pro-social behaviour and that people acquire this strong sense of moral responsibility by engaging in pro-social behaviour [33]. Moral norms help to explain environmentally beneficial behaviours such as energy saving, recycling, transport choices, and purchasing green products [34].

The internal attribution of harmful behaviour often triggers an emotional response known as guilt [35]. Guilt is defined as “a feeling of distressing remorse when actually caused, expected to cause, or associated with a repulsive event” [36] and is an important pro-social emotion because it leads to a feeling of obligation to make amends for the harm caused, and a mismatch between one’s behaviour and social norms can lead to guilt [37]. A mismatch between a person’s behaviour and social norms can lead to feelings of guilt [25].

The function of games is to make people immerse themselves in the activity, and application designers often seek to create designs that increase the frequency of use and make people addicted [38]. Addiction to games may have negative psychological consequences for people [39]. People who play games to the detriment of their other daily schedules or family members who consider playing games to be a form of irresponsibility can cause players to give up playing because of guilt [10].

Generally, people want to see themselves as moral, but they also allow themselves to engage in morally questionable behaviour (such as lying) from time to time. Research has shown that such morally questionable behaviour is more likely to occur when people act in an ethical manner for the first time. Engaging in ethical behaviour beforehand may provide a justification for people to behave less ethically at a later point in time (Meijers et al., 2018). This effect is referred to as the moral licensing effect.

The moral ‘credit’ model is a type of moral permission model in which individuals accumulate credits in a metaphorical moral bank account and then use these credits to buy positive behaviours or offset negative behaviours in order to maintain an overall positive balance in their moral ledger [40].

The permission effect arises because individuals pursue multiple, sometimes conflicting, goals (e.g., advancing their careers or pursuing pro-sociality). For a permission effect to occur, there must be a motivational conflict in the target behaviour (e.g., between self-interest and doing the ‘right’ thing), which manifests as temptation or doubt. In the case of temptation, a person wants to do something personally beneficial (e.g., refuse to help, or cheat in an exam) but is torn by a conflicting motivation (to help, to be an honest person). In such cases, previously positive initial behaviour can allow people to succumb to temptation by increasing the positive side of the moral ledger (moral credit) [41].

As playing games creates feelings of guilt, pro-social behaviour can compensate for the guilt. By incorporating elements of social responsibility into gamification designs, users can engage in pro-social behaviour while playing games. In order to alleviate the feelings of guilt while playing games, individuals will increase their willingness to engage in pro-social behaviour. At the same time, this time, pro-social behaviour will provide the individual with moral permission to play the game afterward, reducing the tendency to withdraw from the game afterward.

In summary, this study proposes that:

**Hypothesis** **1.***Pro-environmental or pro-social games significantly decrease individuals’ game usage compared to general games*.

**Hypothesis** **2.***Guilt mediates the effect of game type on game usage*.

### 2.4. Self-Control

Self-control is defined as taking an action that produces a more positive outcome in the long term than in the short term [25]. Previous research has found a significant correlation between self-control and pro-social behaviour. People with high self-control are more likely to adopt pro-social behavioural strategies rather than selfish ones [42]. Therefore, a gamification design with socially responsible elements would be more likely to increase the amount of play among users with high self-control relative to those with low self-control. In summary, this study proposes that:

**Hypothesis** **3.***Self-control moderates the relationship between guilt and game usage in the gamification design of socially responsible elements; for players with high self-control, the gamification design of socially responsible elements significantly affects players’ game usage; for players with low self-control, the gamification design of socially responsible elements does not have a significant effect on players’ game usage*.

Based on the above assumptions, the model proposed for this study is shown in Figure 1.

## 3. Method and Results

Two studies were used to test the effect of game strategy type on platform game usage. In study 1, we first investigated the famous environmental games through WOM to determine the object of study 1. The survey subjects are mainly college students. Then, we recruited 200 undergraduates from Wuhan University by posting the announcement and finally got 106 valid questionnaires, including 47 females (average age 20) and 59 males (average age 21). In Study 2, we recruited 196 participants from different industries on the questionnaire platform Credamo to conduct our study (mean age 35, of whom 88 were female, 108 were male). Among them, 35 are students, 107 are office workers and 54 are from other industries. The aim of Study 1 was to test Hypothesis 1, Hypothesis 2, and Hypothesis 3 by recruiting participants to complete a questionnaire. The aim of Study 2 was to further test our Hypothesis 1, Hypothesis 2, and Hypothesis 3 by manipulating participants’ feelings of guilt.

### 3.1. Study 1

#### 3.1.1. Materials

We invited 50 random university students at Wuhan University through word of mouth and asked each student to provide the names of one or two environmental games. Thirty-seven of them mentioned Ant Forest in the Alipay software, and 15 mentioned Ant Manor, See Figure 2. Therefore, this paper chooses Ant Forest and Ant Manor inside the Alipay software as the environmental games.

#### 3.1.2. Procedures

We randomly invited 200 students from Wuhan University through poster announcement to participate in the study. When the participants arrive at the study site, they are asked to read a prepared manuscript in front of a computer screen. The content of the manuscript is to inform the participants that they are taking part in a game study. If they agree to participate in the study, the research process will continue, if not, they can leave the study location. Then, the subjects were asked to fill in a questionnaire. The subjects were first asked to recall their experience of playing a game that made them feel guilty. Then, they answered the questions on guilt and game withdrawal. The participant was then asked if they had played Ant Forest or Ant Manor, and if they had, they were then asked to answer the same guilt and game withdrawal questions. If the participant answered that they had not played the game, the survey ended. Finally, subjects completed the self-control questions and demographic information. At the conclusion of the study, participants were informed verbally about the study and urged not to discuss the study with others. After excluding players who did not complete the questionnaire and those who had not played Ant Forest or Ant Manor, 106 valid questionnaires were returned, giving an overall validity rate of 53%, with 59 males (average age 21) and 47 females (average age 20).

The guilt items such as “After playing the game described in the previous question (or while playing the game, or before playing the game) you sometimes feel guilty” and “Sometimes you feel sad that playing this game interferes with your normal study, work, life and social schedule “were measured on a 7-point Likert scale (1 indicating strong disagreement and 7 indicating strong agreement) [43]. Self-control questions such as “I am good at resisting temptation” and “I have a hard time breaking bad habits” were scored on a 5-point Likert scale, with 1 indicating strongly agree and 5 indicating strongly disagree. The rest of the questions were scored on a 7-point Likert scale [44]. The items for each variable are shown in Table 1.

#### 3.1.3. Results

The Cronbach’s alpha coefficient for each variable was measured using SPSS 25.0 software and the results are shown in Table 2.

The results of all the items in this questionnaire are above 0.8, and the questionnaire designed for this study can be considered to have good reliability.

Validated factor analysis was conducted on each variable, and the KMO coefficient was 0.944, and Bartlett’s spherical test was less than 0.05. The total variance explained was greater than 60%, and the factor loadings were greater than 0.6, indicating good construct validity.

A regression analysis of the effect of guilt on game withdrawal in two contexts (environmental games and normal games) was conducted using spss25.0. A logistic regression model was developed with game withdrawal as the dependent variable and guilt as the independent variable. The results showed that guilt had a significant effect on the reduction of game frequency (β = 0.708, *p* < 0.001) and the reduction of game time (β = 0.734, *p* < 0.001) in the normal game context. In the environmental game context, guilt had a significant effect on reduced frequency of play (β = 0.808, *p* < 0.001) and a significant effect on reduced time spent playing (β = 0.812, *p* < 0.001).

A bootstrapping technique with 95% confidence intervals and 5000 samples was used to test for mediation (Hayes 2013, model 15). Results revealed a significant indirect effect of the pro-social game on game withdrawal frequency through guilt, B = 0.1374, [LLCI: −0.9806, ULCI: −0.2809]. Results also revealed a significant indirect effect of the pro-social game on game duration reducing through guilt, B = 0.1374, [LLCI: −0.9806, ULCI: −0.2809]. The mediating effect is significant when self-control is high and insignificant when self-control is low.

### 3.2. Study 2

#### 3.2.1. Procedures

We invited 212 individuals to participate in our survey from the questionnaire platform Creator of Data and Model (Credamo). Credamo is an intelligent, professional research platform that provides a more diverse and representative sample than a typical undergraduate research pool. The platform allows for the recruitment of participants who meet our requirements to participate in our research. Participants who did not pass the attention test and those who did not complete the questionnaire as required were removed, resulting in 196 valid questionnaires, of which 88 were female, and 108 were male, with an average age of 35 years Among them, 35 are students, 107 are office workers, and 54 are from other industries.

In study 2 we manipulated the level of guilt to further test our hypothesis [43]. As with Study 1, subjects first read a prepared manuscript informing them that they were participating in a study about the game. If they agreed to participate, they moved on to the next process, and if they did not agree to participate, they could choose to forgo participation. The subjects first recalled their own experiences of gaming guilt and then answered a questionnaire reporting guilt and the extent to which they had reduced playing this game. Two different sets of brochures from Childreach, a charitable organization, were then read and questions were answered, see Figure 3. Both sets of brochures introduced the subjects to Childreach as a charitable organization that aims to help children in poor places. However, those in the high social responsibility group were shown a page with an introduction to Childreach and a picture in black on white, followed by a description of the poor conditions of the children in the poor areas in which the subjects were placed in the first person. The low social responsibility element group could only see an introduction to Childreach. After reading the brochure, both groups answered the same questionnaire, which tested the subjects’ sense of responsibility for the child and their guilt for not helping. Both groups of subjects were then told that the game they had initially been asked to recall added the feature that by playing the game they could obtain help from the game vendor for Childreach and thus help the child. The subjects were then asked to imagine themselves playing the game and then to fill in a questionnaire to answer how guilty they felt about playing the game and how likely they were to be less likely to play it. After completing these steps, participants were given a detailed description of the entire study and were asked not to discuss it with anyone else.

#### 3.2.2. Results

Manipulation test. The results of the independent samples t-test showed that the guilt of the subjects in the high guilt group (M = 5.85, SE = 0.118) was significantly greater than that of the participants in the low guilt group (M = 5.02, SE = 0.175, *p* < 0.05), indicating successful manipulation.

Game withdrawal. The results of the independent samples t-test showed that the high-guilt game group played less frequently (M = 5.61, SE = 0.124) than participants in the low-guilt group (M = 5.13, SE = 0.174, *p* < 0.05). The reduction in play time for the high-guilt game group (M = 5.63, SE = 0.127) was significantly greater than for participants in the low-guilt group (M = 5.04, SE = 0.180, *p* < 0.05).

Intention of pro-social behaviour. Intention to behave pro-social was significantly higher in the high-guilt group (M = 5.64, SE = 0.129, *p* < 0.05) than in the low-guilt group (M = 5.01, SE = 0.168, *p* < 0.05).

Regression analyses were conducted using the bootstrap technique with game type as the independent variable, game withdrawal as the dependent variable, guilt as the mediator and self-control as the moderator (Hayes, model 15). Results revealed a significant indirect effect of pro-social game on game withdrawal frequency through guilt, B = −0.29, [LLCI: −0.5912, ULCI: −0.0631]. Results also revealed a significant indirect effect of pro-social game on game duration reducing through guilt, B = −0.2436, [LLCI: −0.4780, ULCI: −0.0537]. The mediating effect is significant when self-control is high and insignificant when self-control is low.

## 4. Discussion

Game strategies are widely used by companies to attract users and increase their stickiness. At the same time, the protection of the ecological environment is also an important expression of corporate social responsibility. Through two studies, our paper explores the impact of pro-environmental or pro-social gaming strategies on the use of games on platforms, as well as their underlying mechanisms and boundary conditions. An independent samples t-test of study 2 showed that the high-guilt game group played less frequently (M = 5.61, SE = 0.124) than participants in the low-guilt group (M = 5.13, SE = 0.174, *p* < 0.05). The reduction in play time for the high-guilt game group (M = 5.63, SE = 0.127) was significantly greater than for participants in the low-guilt group (M = 5.04, SE = 0.180, *p* < 0.05). The results support our 1 that environmental or pro-social game strategies significantly reduce individual game use. A regression analyses of study 1 revealed a significant indirect effect of the pro-social game on game withdrawal frequency through guilt, B = 0.1374, [LLCI: −0.9806, ULCI: −0.2809]. Results also revealed a significant indirect effect of the pro-social game on game duration reducing through guilt, B = 0.1374, [LLCI: −0.9806, ULCI: −0.2809]. The mediating effect is significant when self-control is high and insignificant when self-control is low. A regression analyses of study 2 also revealed a significant indirect effect of pro-social game on game withdrawal frequency through guilt, B = −0.29, [LLCI: −0.5912, ULCI: −0.0631]. Results also revealed a significant indirect effect of pro-social game on game duration reducing through guilt, B = −0.2436, [LLCI: −0.4780, ULCI: −0.0537]. The mediating effect is significant when self-control is high and insignificant when self-control is low. The results of Study 1 and study 2 showed that the mediating effect of guilt will be significant when individuals have a high sense of self-control, while the effect of guilt will not be significant when individuals have a low sense of self-control, which supports Hypothesis 2 and Hypothesis 3. Additionally, this result validates the previous hypothesis that individuals with a high sense of self-control will be more concerned with long-term benefits than short-term benefits [25].

In terms of our theoretical contribution, gamification strategies have an important role in promoting environmentally friendly behaviour among users [45]. Most previous studies on platform gamification strategies have considered gamification from the perspective that gamification de facto can attract users to the platform [3,13,46,47]. Mulcahy el. found that gamification has a positive impact on consumers’ knowledge, attitudes and behavioural intentions, and that reward-based game design elements can enhance short-term sustainable behaviour [16]. Several studies have also explored the effects of motivational change and user characteristics on user engagement [48]. Kok et al. found that different games have different effects in different cultures, with games that do not compete with each other being rated more positively in collective cultures [49]. Some studies have also found that gamification designs embedded with environmental claims applications and gamification elements related to environmental sustainability have a positive effect on attitudes towards games [50] and that positive gamification has a positive effect on encouraging environmentally friendly behaviour among employees [51]. Fewer studies have examined the joint effects of the negative psychological effects of gaming and the psychological mechanisms of pro-social behaviour on consumer behaviour. Recent studies have shown that environmental narratives in gaming environments can enhance self-efficacy and thus have a positive impact on environmental behaviour [52]. In line with their findings, our paper builds on this by further investigating the psychological mechanisms inherent in socially responsible games from the perspective of influencing game exit, extending research on platforming strategies, gamification design and pro-social behaviour to demonstrate that socially responsible elements as part of game affordances can increase the long-term use of game strategies. At the same time, our study extends research on the persistence of pro-social behaviour by demonstrating that socially responsible games can reduce game use intentions but increase pro-environmental or pro-social behaviour and reduce game withdrawal behaviour caused by addiction to games.

Limitation and future directions. This paper explores the effects of pro-environmental or pro-social gamification designs on environmentally sustainable behaviour through two studies, while the results support our related hypotheses. Although our hypothesis is based on the findings of previous studies that game mechanics can generate guilt and thus lead to withdrawal behaviour [4], no relevant secondary data were collected. Subsequent studies could further test our hypothesis based on corporate or government data. Additionally, guilt in our study was collected using individual recall, which may bias the measurement of guilt, and subsequent studies could design pro-environmental or pro-social games to investigate individuals’ psychological experiences in real time. Finally, our study only draws conclusions in the context of the Ant Forest or Ant Manor games, and future work is needed to explore how different game types or different designs of pro-environmental or pro-social elements affect different individuals.

## 5. Conclusions

Previous studies have suggested that the mechanism of game addiction can have negative psychological effects on players [39]. Indulging in games will take up time from other activities, or expose players to social pressure or self-blame, leading to guilt [10]. This is consistent with our findings that participants feel guilty when they realize that the game is affecting their daily lives or develop a sense of self-responsibility. Additionally, our results found that pro-environmental or pro-social game use was significantly lower than general game use, which supports our Hypothesis 1. The results also showed that pro-social or pro-environmental games reduced game use through the mediating effect of guilt, supporting Hypothesis 2. Finally, we also found that our findings were significant when individuals had greater self-control, supporting Hypothesis 3.

The results of Study 2 reveal that users are more willing to engage in pro-social behaviour when the pro-social elements of the game activate high levels of guilt. Previous research has shown that games make people feel guilty mainly because the game mechanics make users addicted [38]. People want to get rid of this guilt, which can be compensated for by reducing their personal playtime and by engaging in pro-social behaviour. The guilt associated with the game therefore increases the user’s willingness to adopt socially responsible behaviours, thus promoting the long-term use of environmentally friendly games. Whereas previous research has found that the use of gaming or gamification measures can change users’ short-term environmentally sustainable behaviour [33], our findings suggest that the guilt generated in individuals towards environmental or charitable behaviour can provide some suggestions for the long-term effectiveness of environmentally sustainable behaviour. Future research is needed to further validate the long-term effectiveness of guilt through secondary data or experiments.

Our results also extend the research on corporate social responsibility. While companies are increasingly adopting gaming or gamification measures to fulfil their corporate social responsibility and enhance their reputation. However, research on the effectiveness of gamification has mainly explored the positive effects of gamification, such as satisfying users’ intrinsic motivation and satisfaction [45]. Our research found that environmental or pro-social elements can increase the effectiveness of socially responsible gamification by reducing the guilt of playing games. However, further research is needed on how pro-environmental or pro-social elements can be designed into games or gamification.

Practical implications: Sustainable behaviour in environmental protection is a topic of great concern to businesses and governments. With the advancement of technology, environmental protection is becoming an increasingly pressing issue for mankind. To reduce energy waste, government departments or companies have devised many disruptions to influence people’s daily energy use habits, and combining environmentally friendly behaviour with gamification has proven to be an effective disruption. For example, in the six months after residents played the sustainability game, people used significantly less electricity in their homes [53]. Gamification of environmental knowledge, in a game-based reward mechanism, can enhance sustainable behaviour of residents in the short term [16]. However, research has also found that environmentally sustainable behaviours following gamification interventions are not maintained after a period of time, and that gamified changes in environmental awareness do not translate into long-term environmental behaviour [18]. Our findings found that pro-environmental games with high levels of guilt reduced game addiction and increased the propensity for pro-social behaviour. Although in the short term, there is a transient reduction in play time and play frequency, environmentally sustainable behaviour in play is a long-term process and the reduction in play addiction reduces play withdrawal behaviour brought about by guilt, which is a positive influence on the long-term effects of environmental play. Based on this, enterprises or government departments can balance the relationship between pro-environmental or pro-social elements and game indulgence elements in subsequent environmental gamification strategies, appropriately adjusting the description form or expression of pro-environmental or pro-environmental claims to increase users’ guilt about pro-environmental or pro-social behaviours, while appropriately reducing game elements or mechanisms that bring about personal indulgence to reduce the guilt brought about by game behaviours.

On the other hand, from the perspective of corporate gaming or gamification strategies in general, our research can provide further insights into corporate gaming strategies. Games and gamification designs are one of the means to attract users to platforms [54], and more and more platform companies are embedding games or gamification designs in their platforms [55], but games or gamification designs also have the potential to cause users’ withdrawal behaviour, which can affect companies’ strategies. Companies can incorporate social responsibility elements into their platform games as part of the availability of gamification design to increase the volume of games and user stickiness. Additionally, as different socially responsible element designs may allow consumers to perceive high and low levels of social responsibility, companies should be close to their users when incorporating socially responsible element designs that allow them to perceive high levels of social responsibility.

## Figures and Tables

**Figure 1 ijerph-19-13406-f001:**
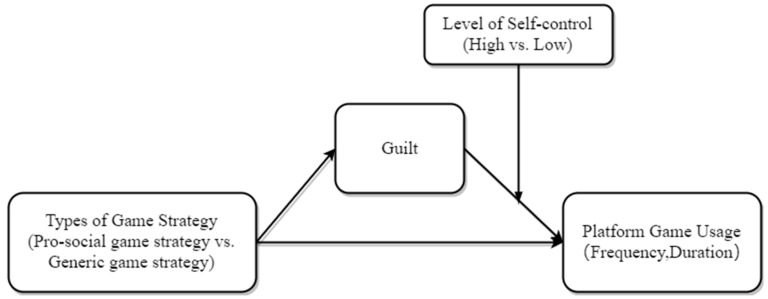
Concept model.

**Figure 2 ijerph-19-13406-f002:**
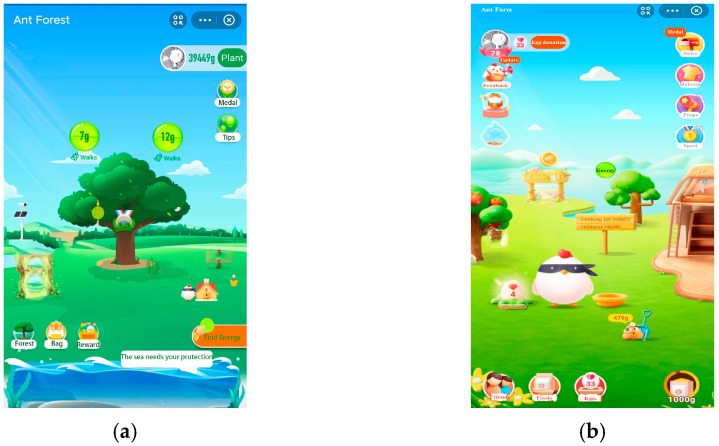
(**a**) Ant Forest; (**b**) Ant Manor.

**Figure 3 ijerph-19-13406-f003:**
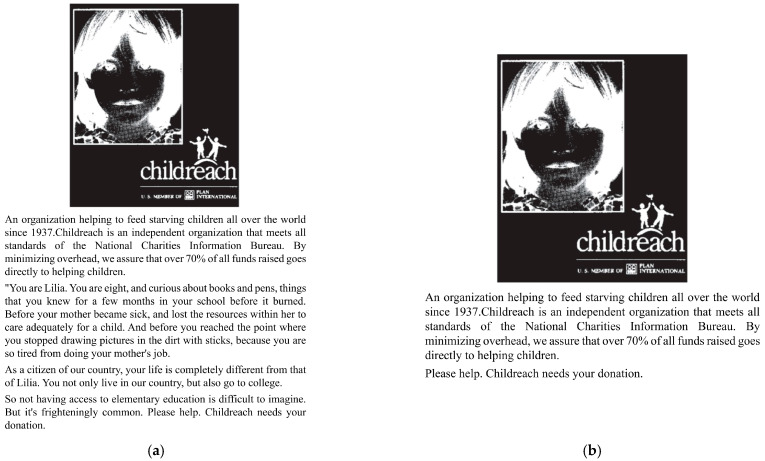
(**a**) High guilt; (**b**) Low guilt.

**Table 1 ijerph-19-13406-t001:** This is a table. Tables should be placed in the main text near to the first time they are cited.

Variable	Item
guilt	After playing the game described in the previous question (or while playing the game, or before playing the game) you sometimes feel guilty
	2. Sometimes you feel sad that playing this game interferes with your normal study, work, life, and social schedule
Self-control	I am good at resisting temptation
	2. I have a hard time breaking bad habits 3. I am lazy
	4. I do certain things that are bad for me, if they are fun 5. I refuse things that are bad for me 6. I wish I had more self-discipline 7. People would say that I have iron self- discipline
	8. Pleasure and fun sometimes keep me from getting work done
	9. I have trouble concentrating
	10. I am able to work effectively toward long-term goals
	11. Sometimes I can’t stop myself from doing something, even if I know it is wrong
	12. I often act without thinking through all the alternatives
Game withdrawal	To what extent do you reduce the frequency of playing games?
	2. To what extent do you reduce the duration of playing games each time?

**Table 2 ijerph-19-13406-t002:** Cronbach’s alpha coefficient for each variable.

The Guilt of Normal Games	The Guilt of Environmental Play	Self-Control
0.837	0.862	0.885

## Data Availability

The data presented in this study are available on request from the corresponding author. The data are not publicly available due to privacy.

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
