# Peer review of "Research on the Impact of Pro-Environment Game and Guilt on Environmentally Sustainable Behaviour"

_ijerph, 2022, doi:10.3390/ijerph192013406_

Round 1
Reviewer 1 Report
I believe the paper's topic is very interesting and will interest the Journal readers. As the authors state, the paper explores the impact of pro-environmental or pro-social gaming strategies on the use of games on platforms and their underlying mechanisms and boundary conditions.
The paper is well written, and the research methodology is sound. The hypotheses are clearly stated, and two studies are used (study 1 and 2) to test the hypotheses. The findings and the conclusions are also stated clearly in the paper and are consistent with the evidence presented.
I only have the following suggestions to make for improving the quality of the paper:
I believe that the goals of the two studies (3.1. Study 1 and Study 2) should be stated before these two sections. How do they differ, and what is intended by these two studies? It is mentioned in Study 2 that the level of guilt is manipulated to further test the hypothesis, but it will be better if the goals are stated before studies 1 and 2.
Furthermore, more information should be given about the Credamo platform (line 286).
Reviewer 2 Report
First of all, thank you for inviting me to review the paper “Research on the Impact of Pro-Environment Game and Guilt on Environmentally Sustainable Behaviour”. Some suggestions and revision recommendations are provided:
Abstract – do include the methodology, instrument, and number of participants (including type of participants eg. students)
Introduction – some of the facts presented in paragraph 1 (lines 23 to 36) should be cited. For instance, the “carbon emissions by 7.92 million tons in three years” this should be cited. Same with other facts mentioned within the introduction.
Similar with this statement “gamification can provide players with a sense of enjoyment, control, competence, and other positive effects, it can also have negative effects”. I assumed a study was done to prove this, a citation is needed. Same with the next statement (including facts from lines 46 to 58).
“This guilt becomes a disincentive to play, which may lead to withdrawal behavior” – citation needed. This seemed to be a major issue of your paper, proper referencing is needed.
The author/s then proceed to introduced game, game strategy, and social responsibility, guilt and ethical “licence” – check spelling.
Recommend to add background literature on playing games create guilt – This seemed to be the main issue and should be explained further.
Would recommend to include how consent to participate where achieved in the procedure section (both parts 3.1.2 and 3.1.1 - which should actually be 3.2.1 – recheck your subsection numbering). How random invitation was done. (snowballing or word of mouth or poster announcement?)
Additionally, study 1 and study 1 should be noted during the start of the method section; eg study design.
Discussion is too weak. This section should include how your study related back to literature review, which is actually also quite weak (missing relevant literature).
Reviewer 3 Report
Very interesting research.
Attention to improving the following:
1. In the introduction, they must justify their statements based on specific data and/or academic citations. They should avoid making claims without clear scientific support.
2. They must clearly cite recent information published in scientific journals and highlight what their research is really contributing with respect to what is already known.
3. Explain more in-depth the recruitment of the participants in the two experiments, and detail more data about their social, economic, and cultural profiles. Keep in mind that behavior is different, it is highly mediated by socioeconomic factors.
4. They must review and order the arguments and contents of the points referred to as DISCUSSION and CONCLUSIONS; the conclusions must be clear and direct based on the hypotheses and on what you have demonstrated and what is new to what is already known. The limitations should be discussed in the DISCUSSION.
Round 2
Reviewer 2 Report
Thank you for inviting me to review the revised version of the paper. Some minor issues are found:
1. the issue of informed consent or the voluntary nature of the participants are still needed - in sections 3.1.2 and 3.2.1
2. the authors need to re-structure the discussion and conclusion sections, as of the moment the discussion is quite weak, only minor discussions were done (besides the last section of the discussion the rests seems to be the conclusion). Would suggest to move parts of the conclusion into the discussion.
Reviewer 3 Report
Thank you very much for the reply. Now the manuscript is much more structured and clear in its arguments.
Author Response
Thank you very much for your professional advice and for taking the time to review our manuscript.